# Brain-Derived Estrogen and Neurological Disorders

**DOI:** 10.3390/biology11121698

**Published:** 2022-11-24

**Authors:** Darrell W. Brann, Yujiao Lu, Jing Wang, Gangadhara R. Sareddy, Uday P. Pratap, Quanguang Zhang, Rajeshwar R. Tekmal, Ratna K. Vadlamudi

**Affiliations:** 1Department of Neurosurgery, Medical College of Georgia, Augusta University, Augusta, GA 30912, USA; 2Department of Obstetrics and Gynecology, University of Texas Health, San Antonio, TX 78229, USA; 3Department of Neurology, Louisiana State University Health Sciences Center, 1501 Kings Highway, Shreveport, LA 71103, USA

**Keywords:** estradiol, neuroestrogen, neurosteroid, aromatase, cerebral ischemia, Alzheimer’s disease, Parkinson’s disease, epilepsy, traumatic brain injury, neurodegeneration

## Abstract

**Simple Summary:**

Classically, 17β-estradiol, the most potent estrogen produced in the body, is considered to be an endocrine hormone, which is produced primarily in the ovaries in females and circulates in the blood to regulate target tissues throughout the body. However, a large body of research has revealed that both astrocytes and neurons create substantial levels of 17β-estradiol in the brains of both males and females. The roles and functions of brain-derived 17β-estradiol have been an area of intense study, and due to advances in techniques and animal models much progress has been made in understanding its actions and importance in the normal and injured/diseased brain. In this review, we examine evidence that brain-derived 17β-estradiol has primarily a beneficial neuroprotective and anti-inflammatory role in various neurological insults and disorders that affect the brain, including stroke, brain injury, Alzheimer’s disease and Parkinson’s disease. We also discuss evidence that brain-derived 17β-estradiol may have detrimental effects in certain situations, especially when over-produced, such as in epilepsy. Finally, we also explore potential future directions for the area and review the distinct functions and mechanisms of action of 17β-estradiol generated from neurons versus astrocytes.

**Abstract:**

Astrocytes and neurons in the male and female brains produce the neurosteroid brain-derived 17β-estradiol (BDE_2_) from androgen precursors. In this review, we discuss evidence that suggest BDE_2_ has a role in a number of neurological conditions, such as focal and global cerebral ischemia, traumatic brain injury, excitotoxicity, epilepsy, Alzheimer’s disease, and Parkinson’s disease. Much of what we have learned about BDE_2_ in neurological disorders has come from use of aromatase inhibitors and global aromatase knockout mice. Recently, our group developed astrocyte- and neuron-specific aromatase knockout mice, which have helped to clarify the precise functions of astrocyte-derived 17β-estradiol (ADE_2_) and neuron-derived 17β-estradiol (NDE_2_) in the brain. The available evidence to date suggests a primarily beneficial role of BDE_2_ in facilitating neuroprotection, synaptic and cognitive preservation, regulation of reactive astrocyte and microglia activation, and anti-inflammatory effects. Most of these beneficial effects appear to be due to ADE_2_, which is induced in most neurological disorders, but there is also recent evidence that NDE_2_ exerts similar beneficial effects. Furthermore, in certain situations, BDE_2_ may also have deleterious effects, as recent evidence suggests its overproduction in epilepsy contributes to seizure induction. In this review, we examine the current state of this quickly developing topic, as well as possible future studies that may be required to provide continuing growth in the field.

## 1. Introduction

17β-Estradiol (E_2_) is a steroid hormone generated from androgen precursors via the action of the aromatase enzyme [1,2] (Figure 1). E_2_ is traditionally thought of as an endocrine hormone and is secreted into the bloodstream primarily by the ovaries in females. However, E_2_ is also a neurosteroid synthesized in the brain of many species examined to date, including birds, rats, mice, amphibians, reptiles, monkeys, and humans [3,4,5,6,7,8]. 

Basally, aromatase is predominantly localized in neurons, but it can be induced in astrocytes following ischemic, excitotoxic, or traumatic brain injury [7,9,10]. The highest levels of aromatase and brain-derived estrogen (BDE_2_) production is in the amygdala, hypothalamus, hippocampus, thalamus, and cortex of most species [3,4,5,6,7,8]. Functionally, BDE_2_ has important roles in both physiological and pathological processes [3,11,12,13]. Neuron-derived E_2_ (NDE_2_), for instance, has been linked to the control of important physiological processes such synaptic plasticity, neurotransmission, reproduction, sociosexual behavior, and memory. [3,13,14,15,16]. As a proposed driver of neural activity, BDE_2_ has also been suggested to play a role in the pathological process of seizure induction in status epilepticus [17]. Furthermore, both NDE_2_ and ADE_2_ have been implicated in numerous studies to exert neuroprotective and anti-inflammatory effects in the brain [3,12,18,19,20]. Studies utilizing global aromatase knockout mice or aromatase inhibitors to restrict BDE_2_ production and analyze the consequences on physiological and pathological processes have contributed significantly to our understanding of the many roles of BDE_2_. Recently, our group developed a forebrain neuron-specific aromatase knockout (FBN-ARO-KO) mouse model that has aromatase depleted specifically in forebrain neurons, and a GFAP-ARO-KO mouse model, which has aromatase deleted specifically in astrocytes to help further elucidate the specific roles of NDE_2_ versus ADE_2_ in the brain [20]. In the sections below, we discuss the role of aromatase and BDE_2_ in various neurological insults and disorders that affect the brain, as well as potential future directions for the field. Since comparatively much more work has been performed in cerebral ischemia than the other neurological disorders, the primary focus of this review is on cerebral ischemia, followed by review of what is known in the other neurological disorders.

## 2. Ischemic Brain Injury

When a blood clot restricts blood supply to a particular area of the brain by blocking a cerebral vessel, it results in focal cerebral ischemia (FCI), often known as stroke [19,21]. In contrary, blood flow is interrupted in the entire brain in global cerebral ischemia (GCI), which can be caused by cardiac arrest, asphyxiation, or hypotensive shock [22,23]. Several studies have shown that both FCI and GCI induce a significant elevation of aromatase expression in the rodent forebrain. For instance, aromatase expression is increased in the penumbra/peri-infarct area of the cerebral cortex and hippocampus at 1 and 8 days after FCI in the rat, but not at 2 h or at 30 d after FCI [24]. Another study showed aromatase elevation in the peri-infarct area at 2 weeks after FCI in the rat, which suggests the elevation of aromatase may extend out to 2 weeks after FCI, but returns to normal levels by 4 weeks [24,25]. Similarly, aromatase and local E_2_ levels are elevated in the hippocampus from 2–7 days following GCI in both sexes of rats and mice [7,19,20,26,27]. In both the FCI and GCI studies, the increase in aromatase expression was shown by double immunohistochemistry to occur in astrocytes [7,19,20,24,25]. Aromatase elevation in the brain after FCI has been suggested to serve a neuroprotective function based on studies using global aromatase knockout mice and aromatase inhibitors [28,29]. These studies revealed that global aromatase knockout or aromatase inhibition results in increased neuronal damage, apoptosis, infarct volume and worse neurological outcome after FCI [28,29]. Interestingly, a recent study found that transplantation of immortalized neural stem cells engineered to over-produce E_2_ in a FCI animal model accelerated and enhanced recovery of sensorimotor function and reduced lesion volume [30]. This finding further supports a neuroprotective role for enhanced local E_2_ levels in the brain. Furthermore, our group identified that central aromatase antisense knockdown in the female rat enhanced neuron damage/loss and increased cognitive dysfunction after GCI, which further supports a neuroprotective role of BDE_2_ in ischemic brain injury [7]. Microglial activation was also increased after GCI in the aromatase antisense knockdown rats, indicating that BDE_2_ may also act to suppress inflammation [7]. Unfortunately, there are no comparative post-mortem studies in humans examining aromatase changes in the brain after FCI or GCI. There is one report that aromatase levels are increased in the serum 24 h after stroke in women, but it is unclear whether serum changes in aromatase reflect changes in brain aromatase expression [31]. Thus, further work is needed to examine aromatase expression in the post-mortem human brain to determine if BDE_2_ could be similarly elevated in humans following ischemic brain injury. 

*Role of Astrocyte-Derived E_2_ (ADE_2_*). Since expression of aromatase is increased in astrocytes following cerebral ischemia it seems likely that this astrocyte-specific elevation of E_2_ plays a major role in the neuroprotective effects observed in the global aromatase knockout mice and aromatase inhibitor/knockdown studies discussed above. Indeed, our group showed that both the intact male and female, and ovariectomized GFAP-ARO-KO mice, which have E_2_ and aromatase depletion in astrocytes, lacked the normal elevation of aromatase and E_2_ in hippocampal astrocytes following GCI, and had increased ischemia-induced neuronal damage and cognitive deficits after GCI [19]. These findings confirm a key role for ADE_2_ in neuroprotection of the hippocampus following ischemic brain injury. Interestingly, our group further showed that hippocampal reactive astrogliosis was significantly decreased after GCI in GFAP-ARO-KO mice, while reactive microgliosis was significantly increased [19]. It has been proposed that astrocytes display different phenotypes including a neuroprotective phenotype (termed “A2” astrocytes) and a neurotoxic phenotype (termed “A1” astrocytes) [32]. A similar classification has been proposed for microglia in which “M1” microglia represent “classically activated microglia” that typically exert proinflammatory actions, while “M2” microglia represent alternative activated microglia that typically exert anti-inflammatory actions”. The A1/A2 and M1/M2 classifications are probably an over-simplification, with multiple subtypes likely to exist [33]. Nevertheless, in our study, we found that hippocampal A2 reactive astrocyte genes were not induced in GFAP-ARO-KO mice following GCI [19]. This finding suggests that ADE_2_ may be necessary for inducing the “neuroprotective” A2 reactive astrocyte phenotype after GCI. Further work revealed that ADE_2_ modulates reactive astrogliosis by regulating pathways known to be critical for induction of reactive astrocytes. For instance, RNA-Sequencing analysis showed attenuation of the IL-6/JAK/ STAT3 signaling pathway in the hippocampus of GFAP-ARO-KO mice after GCI [19], a signaling pathway implicated to play a key role in mediating both ischemic- and injury-induced reactive astrogliosis in the brain [33,34]. It should be noted that reinstating E_2_ levels in the forebrain of GFAP-ARO-KO mice attenuated the enhanced microglial activation and neuronal damage after GCI and reversed the defects in JAK/STAT3 signaling and reactive astrogliosis [19]. These findings support a neuroprotective role for ADE_2_ in the ischemic brain, most likely by facilitating reactive astrogliosis, promoting the neuroprotective A2 reactive astrocyte phenotype, and attenuating microglial activation (Figure 2).

*Role of Neuron-Derived E_2_ (NDE_2_*). In contrast to astrocytes, to our knowledge, there are no reports showing that aromatase is increased in neurons after stroke. Nevertheless, new findings support that NDE_2_ has neuroprotective activity similar to ADE_2_. For instance, early work using aromatase inhibitors or aromatase knockdown in cultured neurons in vitro revealed enhanced neuronal cell death from oxidative stress [35]. In addition, morphine was shown to enhance NDE_2_ release from neurons, and morphine’s neuroprotective effect against Aβ neurotoxicity was lost when aromatase was knocked down in cultured neurons [36]. More recent studies from our group using FBN-ARO-KO mice, in which aromatase and NDE_2_ are specifically depleted in forebrain neurons, has provided important in vivo confirmation of the role of NDE_2_ as a neuroprotectant in the ischemic brain [20]. Both male and female FBN-ARO-KO mice exhibited reduced reactive astrogliosis in the hippocampus after GCI, as well as reduced A2 astrocyte polarization, greater neuronal damage, and significantly enhanced cognitive defects, as compared to controls that lack conditional knockout of aromatase [20]. Interestingly, hippocampal genes involved in reactive astrogliosis, neuroprotection, and neuroinflammation were shown by transcriptome analysis to be significantly down-regulated in FBN-ARO-KO mice as compared to control mice after GCI [20]. This may explain the greater neuronal damage, attenuated reactive astrogliosis, and enhanced cognitive defects in ischemic FBN-ARO-KO mice. Furthermore, hippocampal astrocytes in FBN-ARO-KO mice have attenuated expression of neuroprotective factors after GCI, including BDNF, IGF-1 and aromatase/ADE_2_, as well as the glutamate uptake transporter, GLT-1, which can protect neurons by clearing excess glutamate [20]. This finding indicates that NDE_2_ depletion impairs the neuroprotective function of reactive astrocytes after global ischemic brain injury. Reinstatement of forebrain E_2_ levels reversed the decreased expression of neuroprotective astrocyte factors, as well as the other molecular and functional defects in FBN-ARO-KO mice after GCI [20]. Additional studies using RNAseq and immunohistochemistry revealed that fibroblast growth factor-2 (FGF2) is upregulated in hippocampal neurons in FBN-ARO-KO mice after GCI, while its receptor, FGFR3 is upregulated in hippocampal astrocytes [20]. This is intriguing, as FGF2 is a neuronal factor that acts to *suppress* reactive astrogliosis [37,38]. Further work revealed that blocking FGF2 in FBN-ARO-KO mice attenuated GCI-induced neuronal damage and rescued reactive astrogliosis and expression of the neuroprotective factors, BDNF, aromatase and GLT-1 in astrocytes [20]. As illustrated in Figure 2, it is proposed that NDE_2_ acts to suppress neuronal FGF2 signaling in the hippocampus following GCI, thereby releasing a brake on reactive astrogliosis. The enhanced reactive astrogliosis facilitates release of astrocytic neuroprotective factors, which helps protect hippocampal neurons from ischemic damage. It is also possible that NDE_2_, being a neuroprotective factor itself, may act directly on neighboring neurons to further facilitate neuroprotection in ischemic conditions. 

## 3. Traumatic Brain Injury

A blow to the head can induce traumatic brain injury (TBI), which is a leading cause of mortality and disability in the US. TBI can be due to a penetrating injury such as a gunshot wound or stab wound, or a non-penetrating injury, such as being struck in the head from an assault, fall or car accident. Several groups have examined whether TBI affects aromatase expression in the brain. One of the first studies used a rodent animal model of TBI, which revealed that aromatase is strongly upregulated in astrocytes 8–10 days following a penetrating brain injury [9]. In the study, enhanced aromatase expression in astrocytes was found in all brain regions of rats and mice where a penetrating brain injury was induced in, including the hippocampus, hypothalamus, corpus callosum, cortex, striatum and thalamus [9]. Morphologically, the astrocytes that expressed aromatase exhibited a hypertrophic appearance typical of reactive astrocytes. Local E_2_ was not examined in this study, but subsequent work in zebra finches confirmed upregulation of both aromatase expression in astrocytes and local E_2_ levels beginning as early as 4–6 h after a penetrating brain injury [39,40,41]. At least one study reported that the increased aromatase expression in astrocytes could be maintained out to 6 weeks after brain injury [42]. Furthermore, a study in humans reported that higher cerebrospinal fluid E_2_/testosterone ratio was associated with better outcome [43]. The rapid increase in aromatase expression and local E_2_ after a penetrating brain injury in zebra finch was preceded by an even earlier increase in inflammatory mediators such as interleukin-1beta (IL-1β), interleukin-6 (IL-6) [44] and prostaglandin E_2_ (PGE_2_) [45,46], which are suggested to help induce the enhanced astrocyte aromatase expression and local E_2_ levels after TBI. In support of this contention, induction of inflammation in the brain by administration of lipopolysaccharide or phytohemagglutinin led to strong induction of aromatase in the uninjured zebra finch brain [47,48]. Furthermore, central administration of indomethacin attenuated the increase of aromatase and BDE_2_ in the injured zebra finch brain [45]. Interestingly, elevated pressure has also been shown to elevate aromatase expression and activity in glioma cells in vitro, which could suggest that astrocytes detect increased pressure after TBI to facilitate increased aromatase and ADE_2_ levels [49]. Additional work in mice showed that systemic treatment with an aromatase inhibitor exacerbated neurological deficits in mice at 24 and 72 h after TBI, but due to the systemic nature of the aromatase administration it is unclear whether this effect is due to inhibition of BDE_2_ or gonadal estrogen [50]. However, central aromatase inhibitor treatment in zebra finches with a penetrating head injury was associated with a significant increase in both the lesion size and number of apoptotic nuclei [42,51], effects that could be rescued by E_2_ replacement [52], which suggests that ADE_2_ does exert neuroprotection in the TBI brain. Currently, it is not clear if NDE_2_ has a similar neuroprotective role in TBI as observed after GCI. It would thus be illuminating to utilize the FBN-ARO-KO and GFAP-ARO-KO mice to further delineate the specific roles of ADE_2_ and NDE_2_ in TBI. 

## 4. Excitotoxic Brain Injury and Epileptic Seizures

Excitotoxicity is a term for damage to cells caused by over-activation of glutamate receptors due to over-abundant release of glutamate [53]. Excitotoxicity plays a key role in seizure-induced and ischemic-induced neuronal damage as well as other neurodegenerative disorders such as Parkinson’s disease (PD), Huntington’s disease and Alzheimer’s disease (AD) [53,54]. The excitotoxic glutamate receptor agonist, kainic acid is commonly used in animal models to induce excitotoxicity and seizures. Studies in these animal models have shown that kainic acid induces significant neuronal loss in the hippocampus in rodents which is associated with a robust increase of aromatase expression in astrocytes but not in neurons (1). Administration of kainic acid also enhanced the synthesis of E_2_ in the rat hippocampus [17]. Global aromatase knockout mice have been shown to exhibit significant neuron loss in the hippocampus after low dose treatment with another neurotoxic glutamate receptor agonist, domoic acid, while wild type mice displayed no neuron loss [55]. This result implies that the loss of BDE_2_ and aromatase increases the susceptibility of hippocampal neurons to excitotoxic/neurotoxic damage. In further support of a neuroprotective role of aromatase and BDE_2_ in neurotoxicity, methyl-mercury-induced neurotoxicity in male rat hippocampal slices was enhanced by administration of the aromatase inhibitor letrozole, and this effect could be rescued by E_2_ replacement [56]. Kainic acid administration also is a well-known model of status epilepticus. Interestingly, synthesis of BDE_2_ in the rat hippocampus is robustly increased during status epilepticus [17]. Furthermore, a role for BDE_2_ in inducing seizure activity is supported by the finding that intra-hippocampal aromatase inhibition suppresses kainic acid-induced electrophysiological and behavioral seizures [17]. Clinical case reports also suggest that adding an aromatase inhibitor enhances seizure control in humans [55,57,58]. Thus, targeting over-production of BDE_2_ via administration of aromatase inhibitors may have therapeutic efficacy in epilepsy. Further work is needed in this interesting area.

## 5. Alzheimer’s Disease (AD)

Numerous studies have shown that single nucleotide polymorphisms in the aromatase gene are linked to increased susceptibility to AD, either alone or in combination with other risk factor genes [59,60,61,62,63]. Furthermore, post-mortem studies have found aromatase expression to be altered in the AD brain. For instance, reduced aromatase expression and E_2_ levels were reported in the frontal cortex and cerebellum of AD subjects, which correlated with enhanced amyloid plaque density in the AD cortex [64]. In contrast, other studies found that aromatase immunoreactivity is increased in the hippocampal CA4 region [65] of AD patients, and in prefrontal cortical astrocytes of the late-stage human AD brain [66]. It is not clear why some reports show a decrease while others find an increase in aromatase expression in the AD brain. It could be due to different areas examined and/or different stages of AD progression being assessed. Nevertheless, in support of a potential role for BDE_2_ in regulating plaque formation, global aromatase knockout mice that overexpress amyloid precursor protein (APP) had increased expression of β-site amyloid precursor protein cleaving enzyme 1 (BACE1), the enzyme responsible for beta-amyloid expression, as well as temporally accelerated and increased beta-amyloid deposition, and decreased beta-amyloid clearance by microglia [64]. Furthermore, some evidence suggests that BDE_2_ may protect neuronal connectivity and mitochondrial function from beta-amyloid-induced defects as aromatase inhibition led to enhanced defects in hippocampal mitochondrial and dendritic spines impairments induced by beta-amyloid [67]. Furthermore, exogenous treatment with an estrogen receptor-β agonist was able to rescue defects in mitochondria induced by beta-amyloid in rat hippocampal neurons in vitro [68]. To provide further insights into the cell-specific roles of ADE_2_ and NDE_2_ in AD, future studies are needed using neuron- and astrocyte-specific aromatase knockout mouse models.

## 6. Parkinson’s Disease (PD)

Few studies have looked into how aromatase and BDE_2_ are involved in PD. Examination of global aromatase knockout mice revealed that there was decreased integrity of tyrosine hydroxylase-positive neurons in the substantia nigra and dopamine transporter innervation of the caudate putamen, as well as an enhanced vulnerability to MPTP-induced nigrostriatal damage in the aromatase knockout mice [69]. Furthermore, central aromatase inhibitor administration in a 6-hydroxydopamine-lesioned rat model of PD was found to enhance striatal lesions induced by 6-hydroxydopamine [70]. This finding indicates that BDE_2_ is neuroprotective in the striatum and thus may help protect against neurodegeneration in PD. Further studies are needed to fully address this possibility. 

## 7. Summary and Future Directions

There is mounting evidence that BDE_2_ contributes to a number of neurological insults and disorders. Beneficial roles include neuroprotection, synapse and cognitive preservation, regulation of glial activation and function, as well as anti-inflammatory actions. In contrast, a deleterious role for BDE_2_ has recently been suggested in epilepsy, where it appears to help facilitate induction of seizure activity. Since BDE_2_ is increased in astrocytes in most neurological disorders, it is suggested that ADE_2_ plays an important role in mediating the beneficial effects. However, work using FBN-ARO-KO mice also supports a role for NDE_2_, at least in global cerebral ischemia to mediate these important beneficial effects. While much work has been completed on the role of BDE_2_ in cerebral ischemia and TBI, comparatively much less work has been performed in other neurological disorders such as AD, PD and epilepsy. Therefore, future studies are needed to address this deficit. In addition, utilizing the FBN-ARO-KO and GFAP-ARO-KO mice in disorders other than global cerebral ischemia could also help to further delineate the role of ADE_2_ and NDE_2_ in various aspects of these other neurological disorders. Finally, studies to enhance understanding of how brain aromatase is regulated in astrocytes and neurons are needed, as these could lead to new therapies which could enhance BDE_2_ beneficial effects in neurological disorders. 

## Figures and Tables

**Figure 1 biology-11-01698-f001:**
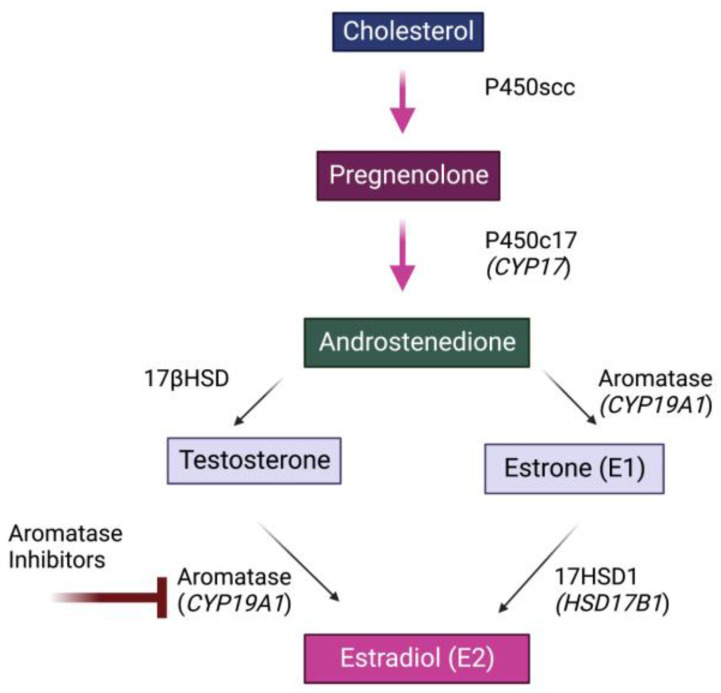
**Simplified Biosynthetic Pathway for Estrogens.** Estrogen synthesis begins with conversion of cholesterol to pregnenolone, which through a series of steps is converted into androstenedione, testosterone and estrone (E_1_). Testosterone is then converted into 17β-estradiol (E_2_) through the action of aromatase (CYP19A1). As also shown, aromatase can be inhibited by various aromatase inhibitors.

**Figure 2 biology-11-01698-f002:**
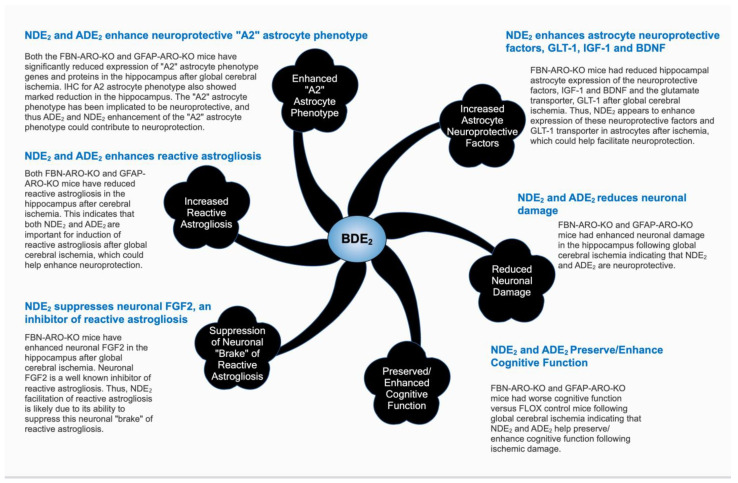
**Proposed Mechanisms Underlying Astrocyte-Derived Estrogen (ADE_2_) and Neuron-Derived Estrogen (NDE_2_) Neuroprotection in Global Cerebral Ischemia.** See text for description.

## Data Availability

Not applicable.

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
