# Peer review of "Brain-Derived Estrogen and Neurological Disorders"

_biology, 2022, doi:10.3390/biology11121698_

Round 1

Reviewer 1 Report

This is an excellent review paper addressing a relevant topic to which the authors have made highly significant contributions: the role of brain-derived estradiol under brain pathological conditions. The paper is well organized, covers the relevant literature and presents a complete, critical, and helpful update of the research advances in the field. I cannot find suggestions to improve its content or presentation.

Author Response

Thanks for the positive comments for our review.

Reviewer 2 Report

In this review article, the authors investigated the role of brain-derived estrogen in several neurological diseases. The manuscript is reasonable and their results seem reliable and robust. However, there are some critical clarifications needed in the writing to ensure complete transparency and accuracy. Some comments are below and would have to be clearly addressed to warrant publication. 

What is meant by "M1/M2 classification of microglia"?

In lines 137 and 138, the authors wrote "FLOX control". What is FLOX Control? Explain clearly

An abbreviations section should be added to explain the abbreviations used in the text. First, explain each item and then write it abbreviate.

Why did you only explain BDE2 in general and not about NDE2 and ADE2 in PD and Excitotoxic Brain Injury?

What is meant by "BACE" in row 231?

Is "GFAP-ARO-KO" really an abbreviation of "astrocyte-specific aromatase knockout"?

Author Response

In this review article, the authors investigated the role of brain-derived estrogen in several neurological diseases. The manuscript is reasonable and their results seem reliable and robust. However, there are some critical clarifications needed in the writing to ensure complete transparency and accuracy. Some comments are below and would have to be clearly addressed to warrant publication. 

What is meant by "M1/M2 classification of microglia"?

We have reworded this section as follows:  “A similar classification has been proposed for microglia in which “M1” microglia represent “classically activated microglia” that typically exert proinflammatory actions, while “M2” microglia represent alternative activated microglia that are typically exert anti-inflammatory actions” The A1/A2 and M1/M2 classifications are likely an over-simplification, with multiple subtypes likely to exist.” (See Page 4, Lines 5-10, revised manuscript).

In lines 137 and 138, the authors wrote "FLOX control". What is FLOX Control? Explain clearly

An abbreviations section should be added to explain the abbreviations used in the text. First, explain each item and then write it abbreviate.

We have replaced “FLOX control” with “controls that lack conditional knockout of aromatase”. (See Page 5, Lines 10-11, revised manuscript). As suggested by reviewer, an abbreviations section has been added. (See Page 1, Lines 44-46, Page 2, Lines 1-3, revised manuscript).

Why did you only explain BDE2 in general and not about NDE2 and ADE2 in PD and Excitotoxic Brain Injury?

Unfortunately, there have been no studies as yet using the neuron-specific and astrocyte-specific aromatase knockout mice in AD and PD. Since aromatase inhibitors block both neuronal and astrocyte-derived aromatase, it is difficult to definitively determine the specific contributions of neuron-derived estrogen or astrocyte-derived in the inhibitor studies in AD and PD. Therefore, we use the general term BDE2.

What is meant by "BACE" in row 231? Sorry for the typing error and confusion. “BACE” has been corrected as “BACE1”.

BACE1 stands for “β-site amyloid precursor protein cleaving enzyme 1”. We have clarified this in the text. (See Page 7, Lines 9-10, revised manuscript).

Is "GFAP-ARO-KO" really an abbreviation of "astrocyte-specific aromatase knockout"?

We have reworded this sentence to say: “GFAP-ARO-KO mouse model, which has aromatase deleted specifically in astrocytes”. (See Page 3, Lines 4-5, revised manuscript).

Reviewer 3 Report

This is an important and timely paper where the authors discuss the importance of brain-derived estrogen and neurological disorders. The authors should be applauded for this review. Overall, the paper is well written but there are some concerns that if addressed will significantly improve the paper:

1. Organization. The first section ends with the sentence, "... ... we discuss the role of aromatase and BDE2 in various neurological disorders that affect the brain as well as potential future directions for the field." The authors proceed to have the following major sub-headings - Ichemic Brain Injury, Traumatic brain injury, Excitotoxic brain injury and epileptic seizures, AD and PD. IBI and TBI are not disorders. When experienced leads to the development of numerous disorders. The subheadings should be reorganized and/or the titles should be reworded. The opening section should also be reframed to set the stage for the rest of the paper. 

One suggestion is to focus the review on stroke, which is the body of the authors' work and draw from other examples like in TBI, AD and PD. 

2. The authors address the effects of IBI or TBI on aromatase as an up or down expression. However, the sequalae of events subsequent to the experience of TBI as with the development of disorders such as AD or PD are not binary. The timing of expression should be discussed in context of outcomes. For example, many of the bird TBI studies tend to be chronic in contrast to the rodent study. Is the idea of aromatase upregulation an initial reaction to form a neuroprotective role? What happens to the initial and more chronic regulation? 

3. There are several papers that are not cited that would better support the views of the authors. These include work from the laboratories of Shafer, Wagner and Simpkins for TBI studies. The same is true for PD and AD discussions. 

Author Response

Reviewer 3

This is an important and timely paper where the authors discuss the importance of brain-derived estrogen and neurological disorders. The authors should be applauded for this review. Overall, the paper is well written but there are some concerns that if addressed will significantly improve the paper:

  1. Organization. The first section ends with the sentence, "... ... we discuss the role of aromatase and BDE2 in various neurological disorders that affect the brain as well as potential future directions for the field." The authors proceed to have the following major sub-headings - Ichemic Brain Injury, Traumatic brain injury, Excitotoxic brain injury and epileptic seizures, AD and PD. IBI and TBI are not disorders. When experienced leads to the development of numerous disorders. The subheadings should be reorganized and/or the titles should be reworded. The opening section should also be reframed to set the stage for the rest of the paper. 

One suggestion is to focus the review on stroke, which is the body of the authors' work and draw from other examples like in TBI, AD and PD. 

Thanks. It should be noted that in January 2022 stroke was reclassified as a neurological disorder in the new International Classification of Diseases 11 (ICD-11). Nevertheless, we have modified the last sentence in the “Introduction” to read:

“In the sections below, we discuss the role of aromatase and BDE2 in various neurological insults and disorders that affect the brain as well as potential future directions for the field” (See Page 3, Lines 6-8, revised manuscript). (Italics added to highlight the exact change in wording).

In addition, we have adopted the excellent suggestion of the reviewer and added a new sentence at the end of the “Introduction” clarifying that the focus of the review is on cerebral ischemia and explaining why:

“Since comparatively, much more work has been performed in cerebral ischemia than the other neurological disorders, the primary focus of this review is on cerebral ischemia, followed by a review of what is known in the other neurological disorders.” (See Page 3, Lines 8-11, revised manuscript).

We feel that with these modifications/clarifications on the scope of the review, it is not necessary to change the organization of the review.

  1. The authors address the effects of IBI or TBI on aromatase as an up or down expression. However, the sequalae of events subsequent to the experience of TBI as with the development of disorders such as AD or PD are not binary. The timing of expression should be discussed in context of outcomes. For example, many of the bird TBI studies tend to be chronic in contrast to the rodent study. Is the idea of aromatase upregulation an initial reaction to form a neuroprotective role? What happens to the initial and more chronic regulation?

Where not indicated, we have now added the timing in the revised manuscript. The timing of aromatase elevation is discussed for FCI and GCI in Page 3, Lines 18-26, revised manuscript. The elevation of aromatase after FCI and GCI is proposed to be neuroprotective and the evidence for this is discussed in Page 3, Lines 26-39, 46-54, Page 4, Lines 10-23, revised manuscript. For TBI, the elevation of aromatase is shown to occur as early as 4-6hrs after TBI, and some studies show the elevation can last for 6 weeks. We have added this temporal data in Page 5, Lines 47-53, revised manuscript. The early acute aromatase elevation is proposed to be induced by an increase in proinflammatory factors that precede the aromatase elevation, and the acute aromatase elevation is suggested to be neuroprotective based on the results of aromatase inhibitor studies in birds (Page 5 Line 54, Page 6, Lines 1-17, revised manuscript). To our knowledge, the role and importance of the chronic elevation of aromatase in birds is currently unknown. Interestingly, it correlates with chronic glial activation after brain injury in birds and may also exert a long-term neuroprotective or anti-inflammatory effect, but this remains to be determined.

3. There are several papers that are not cited that would better support the views of the authors. These include work from the laboratories of Shafer, Wagner and Simpkins for TBI studies. The same is true for PD and AD discussions. 

As recommended by the reviewer, we have now cited and discussed findings from several additional papers in the text (see Page 5, Lines 50-53, Page 6 Lines 7-13, Page 7, Lines 14-15, revised manuscript, new references #42, 43, 49, 50, and 68). 

Round 2

Reviewer 3 Report

The authors have responded to my queries. I thank the authors for responding appropriately and concisely. 

In the future, the authors may want to advance this important work on aromatase by addressing the differences in mild and severe TBI, as well as the intersection of AD-symptoms and TBI.